# State of the Art in Artificial Intelligence and Radiomics in Hepatocellular Carcinoma

**DOI:** 10.3390/diagnostics11071194

**Published:** 2021-06-30

**Authors:** Anna Castaldo, Davide Raffaele De Lucia, Giuseppe Pontillo, Marco Gatti, Sirio Cocozza, Lorenzo Ugga, Renato Cuocolo

**Affiliations:** 1Department of Advanced Biomedical Sciences, University of Naples “Federico II”, 80131 Naples, Italy; annacastaldo1202@gmail.com (A.C.); dav.delucia@gmail.com (D.R.D.L.); giuseppe.pon@gmail.com (G.P.); sirio.cocozza@unina.it (S.C.); lorenzo.ugga@unina.it (L.U.); 2Radiology Unit, Department of Surgical Sciences, University of Turin, 10124 Turin, Italy; marcogatti17@gmail.com; 3Department of Clinical Medicine and Surgery, University of Naples “Federico II”, 80131 Naples, Italy

**Keywords:** hepatocellular carcinoma, imaging, radiomics, machine learning, deep learning

## Abstract

The most common liver malignancy is hepatocellular carcinoma (HCC), which is also associated with high mortality. Often HCC develops in a chronic liver disease setting, and early diagnosis as well as accurate screening of high-risk patients is crucial for appropriate and effective management of these patients. While imaging characteristics of HCC are well-defined in the diagnostic phase, challenging cases still occur, and current prognostic and predictive models are limited in their accuracy. Radiomics and machine learning (ML) offer new tools to address these issues and may lead to scientific breakthroughs with the potential to impact clinical practice and improve patient outcomes. In this review, we will present an overview of these technologies in the setting of HCC imaging across different modalities and a range of applications. These include lesion segmentation, diagnosis, prognostic modeling and prediction of treatment response. Finally, limitations preventing clinical application of radiomics and ML at the present time are discussed, together with necessary future developments to bring the field forward and outside of a purely academic endeavor.

## 1. Background

Hepatocellular carcinoma (HCC) represents the main primary liver cancer, ranking sixth for incidence and representing the fourth most frequent cause of cancer-related mortality, with a median survival of approximately 6 to 20 months. The lion’s share of HCCs occur in a cirrhotic background or patients suffering from chronic liver diseases, usually due to hepatitis B or C virus infection, alcohol abuse or nonalcoholic steatohepatitis [1]. Screening and surveillance of patients at risk can lead to early diagnosis, improved curative therapeutical chances and increased survival. In this view, medical imaging has a central role and guidelines underline the use of ultrasound (US) as the first-level technique, able to supervise healthy and cirrhotic patients [2]. To characterize suspicious liver nodules, contrast-enhanced computed tomography (CECT), dynamic contrast-enhanced magnetic resonance imaging (DCE-MRI) or contrast-enhanced US (CEUS) can be employed [3,4]. A distinctive pattern of hypervascularization followed by wash-out is among the characteristic HCC features included in the Liver Imaging Reporting and Data System (LI-RADS) guidelines, along with size, dimensional increase over time and capsule presence [5]. The LI-RADS criteria achieved a sensitivity of 86% and specificity of 85% for identifying HCC when lesions received a score of at least three out of five [6]. In uncertain cases, biopsy may be required for the final diagnosis. The most commonly used HCC staging system is the one proposed by the Barcelona Clinic Liver Cancer (BCLC) [7]. It classifies patients as being in one of five stages relying on performance status, tumor stage and liver function. BLCL early-stage patients can undergo curative treatments, such as ablation, resection or transplantation, while intermediate and advanced stage ones may benefit from palliative cures, including systemic and trans-vascular therapies [1].

Despite progress in both diagnosis and therapy, HCC prognosis remains generally poor, also due to tumor heterogeneity and biological variability between patients. In this setting, several artificial intelligence (AI) technologies are emerging in recent years as very promising in genetic and histological characterization of HCC from medical imaging. AI and machine learning (ML) are able to build quantitative models from radiological exams, aiming to predict the outcomes to different clinical problems [8,9]. In this review, we will provide an overview of ML principles and of recent studies which have applied AI algorithms to HCC imaging for lesion segmentation, diagnosis, grading, prognosis and treatment response prediction.

## 2. Radiomics, Machine Learning and Deep Learning

Recent technological developments and research breakthroughs have led to a wider introduction of advanced techniques for data and image analysis, aiming to surpass the limitations of current clinical practice. This is reflected by the growing number of radiomics and ML studies that have been published across medicine, and in oncology and radiology specifically, sometimes showing results that are competitive or surpass expert radiologists [10,11,12,13,14,15].

The term “radiomics” was popularized in 2012, representing a collection of techniques to extract a large number of quantitative features from medical images, opening the door for ML analysis.

ML consists of algorithms developed to analyze preformed, usually large, datasets. These models have the potential to improve over time by incorporating additional training data and tuning the algorithm parameters to make their outputs more accurate on new cases (i.e., generalizable). Based on how the training process is conducted, different types of ML can be identified [16]:Supervised learning: the training is performed on a set of data labeled in relation to the class of interest, and this information is available to the algorithm while building the model;Unsupervised learning: the training process is performed without availability of class labels for the algorithm, usually producing data clusters that require post hoc interpretation (Figure 1);Reinforcement learning: training based on positive and/or negative feedback loops, not commonly employed in medicine.

Deep learning (DL) is a subset of ML models based on a neural network (NN) structure, originally inspired by the human brain. One of its advantages is the possibility to integrate feature extraction from input data (e.g., medical images) within the model-building process [17]. NNs are mathematic architectures organized in nodes (i.e., “neurons”) organized in layers and interconnected to each other [18].

Today DL is often employed in medical image analysis but its complexity limits model interpretability, which is particularly troubling for the field of healthcare. This has been defined as the “black-box” nature of NNs and requires further research to improve understanding and favor clinical implementation [18].

## 3. Lesion Segmentation

Segmentation is the task of producing annotations delineating target organs or structures (e.g., lesions) of interest. The ability to obtain accurate and reproducible masks from medical images has the potential to allow more reliable analytic applications, such as healthy organ volume and lesion extension assessment prior to treatment planning, identification of prognostic biomarkers, lesions classification and staging.

While manual segmentation performed by experts is still considered the gold standard, it is very time-consuming, exhibits limitations due to inter-reader reproducibility, expertise level and software/hardware employed [19]. AI shows its most promising developments especially in fully automated segmentation. Advanced computational technologies may reduce errors, optimizing time and increase standardization of methods. Liver segmentation, of both healthy parenchyma and tumoral tissue, presents additional complications linked to the imaging characteristics of normal and pathological tissue, size and morphology inter-subject variability, para-physiological and post-treatment modifications as well as imaging protocol and equipment used [20].

In this setting, recent studies on ML have focused on DL, which showed improved results and led to major breakthroughs in medical image analysis. In particular, CNNs have reached significant accuracy in solving computer vision tasks such as object recognition, classification and segmentation, often outcompeting previous state-of-the art methods [21,22,23]. Christ et al. [20] first developed liver and tumor segmentation models using cascaded CNNs, based on a U-Net Architecture [24], tested on the 3DIRCADb dataset and a clinical CT dataset from multiple scanners and centers, reaching DICE scores of 94.3% and 91%. They also demonstrated the versatility of their method by testing it on a novel diffusion-weighted MRI dataset and a large multi-centric CT dataset, again obtaining promising results (DICE score 87–88%). Similarly, Ouhmich et al. [25] used a cascaded CNN based on the U-Net architecture for segmentation of healthy and cancerous liver tissues, discriminating normal parenchyma, active HCC and necrotic tumoral tissue. They confirmed the validity of this approach on an in-house database containing ground truth 2D annotations for each class, on a multi-phasic CT protocol (DICE score of 90.5%, 59.6% and 75.8%, respectively). Zhang et al. [26] investigated the use of an automated context-based NN with a multi-phase training framework to classify different types of liver tissue on 3D multi-parametric MRI in patients with HCC. This solution achieved results comparable with the benchmark method without using manual inputs.

Further advancements on this topic came from Han et al. [27], within the Liver Tumor Segmentation Challenge [28]. Their CNN method, working in 2.5d, placed first in the competition. Wardhana et al. [29] also delved into this approach, evaluating how parameter variations within the 2.5d model could increase its performance. An alternative approach has been proposed by Conze et al. [30], who implemented a semi-supervised scale-adaptive supervoxel-based random forest method, based on annotations by expert practitioners on dynamic contrast-enhanced CT scans. Chih-Yu Hsu et al. [31] proposed a novel active contour model, called Poisson gradient vector flow, based on a genetic algorithm. This solution automatically detects the contours of liver in PET images.

In Table 1, a summary of the reviewed papers is presented.

## 4. Diagnosis

In clinical practice, the main feature to characterize HCC is its pattern of arterial phase hyperenhancement and venous or delayed phase washout on contrast-enhanced CT and/or MRI [1]. Even though this behavior is the hallmark of HCC, other hypervascular benign and malignant lesions can pose differential diagnosis problems [9]. Guidelines establish using US in the surveillance of healthy and cirrhotic patients with nodules smaller than one centimeter, while CT and MRI can be used to characterize suspicious lesions. For lesions whose diagnosis remains unclear, biopsy is required [1]. Computer-aided diagnosis/detection based on ML is gaining growing attention in medical practice due to the promise of increasing diagnostic performance [32]. DL tools may therefore aid in avoiding invasive procedures [17]. An overview of the considered studies is summarized in Table 2.

### 4.1. US

While US is the exam of choice in patient screening and lesion surveillance, it is operator-depending and has poor inter-individual reproducibility [48]. Moreover, it is challenging to distinguish between cirrhotic regenerative changes and HCC, as these show similar features on US.

Bharti et al. [33] proposed an ML model to discern four classes of liver images on US, namely normal liver, chronic liver disease, cirrhosis and HCC, through a selected set of “handcrafted” texture features. These were obtained using ranklet, gray-level difference matrix and gray-level co-occurrence matrix methods and then used as input for an ensemble classifier, a voting algorithm for combining three classifiers, k–nearest neighbor, support vector machine and rotation forest. This pipeline showed an accuracy of 96.6%. Schmauch et al. [34] designed a DL system able to detect and classify space-occupying lesions in the liver as benign or malignant [34]. After supervised training on a database of 367 images paired with radiological reports, the resulting model detected and characterized lesions with a mean area under the receiver operating characteristic curve (AUC) of 0.93 and 0.916, respectively for each class. Hassan et al. [35] used a stacked sparse auto-encoder system to detect HCC, hemangioma and liver cysts on US images. This group employed a four-step framework: first distinguishing images of interest from the background; segmentation by level set method and fuzzy c-means clustering algorithm; then, a stacked sparse auto-encoder identified latent features; in the end, a SoftMax layer was used to diagnose different focal liver diseases showing a sensitivity and specificity of 98% and 95.7%, respectively.

Interesting results were also reported, using CEUS, by Guo et al. [36]. A two-stage multi-view ML framework was developed, based on lesion behavior in three phases (arterial, portal and late) of CEUS. This was able to discriminate benign and malignant liver tumors. They employed a canonical correlation analysis to identify six-view features used by a multiple kernel learning classification algorithm. The final accuracy, sensitivity and specificity were 90.41% ± 5.80%, 93.56% ± 5.90%, and 86.89% ± 9.38%, respectively.

### 4.2. CT

CE CT is the most used imaging modality to characterize liver lesions, thanks to its availability and accuracy. LI-RADS, in cirrhotic patients, can aid diagnosis, but not without limitations [1]. In recent years, several ML approaches have been developed to optimize HCC CT imaging, decrease time to diagnosis and avoid invasive procedures.

Mokrane et al. [37] investigated a DL technique to classify hepatic nodules as HCC or non-HCC. Their test set was a retrospectively collected database of 178 patients with cirrhosis and liver nodules in whom the LI-RADS criteria were unable to obtain a diagnosis, thereby necessitating biopsy; 77% proved to be malignant on biopsy. Nodules were segmented on each phase of triphasic CT scans, and 12 sets of quantitative imaging features, reflecting both lesion behavior on each phase as well as changes between two phases, were extracted. The final model’s AUC was 0.66, sensitivity 0.70 and specificity 0.54.

In another retrospective study [38], a three-layer ANN was trained on over 55,000 images to differentiate liver masses on contrast-enhanced CT into five categories: A, classic HCC; B, malignant tumors apart from HCC (cholangiocarcinoma or metastasis); C, indeterminate masses, dysplastic nodules or early HCC and benign masses other than cysts or hemangiomas; D, hemangiomas; E, cysts. The ANN obtained high accuracy, especially for the differentiation between categories A-B from C-D with an AUC of 0.92. Raman et al. [39] used CT texture analysis on arterial-phase images of FNHs, hepatic adenomas, HCCs and normal liver parenchyma. A commercial software (TexRAD) extracted 32 features, used to build a predictive classification model based on a random forest algorithm. Its accuracy in discerning lesion types was 90%, better than two radiologists (72.2% and 65.6%). Nayak et al. [32] developed a computer-aided diagnosis system, using multi-phase CT, for detection of cirrhosis and HCC. They employed semi-automatic 3d segmentation, based on modified region-growing, and a support vector machine. The tool reached a DICE score of 90% for healthy liver, 86% for cirrhosis and 81% for HCC. Tumor load represents a crucial nodule feature to differentiate benign and malignant nodules, and Vivanti et al. [40] reported a CNN-based detection method to automatically identify lesion recurrence, based on lesion initial appearance on CT, the quantification of the tumor load at baseline and during the follow-up. This technique showed a high identification rate for tumor recurrence, with an accuracy of 86%.

Other works demonstrated how AI-based systems could allow an optimization of CT protocols by reducing the number of phases needed to characterize nodules. Shi et al. [41] compared results obtained by DL, based on three dense CNNs, to differentiate HCC from other focal liver lesions on a four-phase (model A), three-phase without portal-venous phase (model B) and three-phase images without pre-contrast phase (model C) CT protocol. In their results, model B, using a three-phase CT protocol without pre-contrast images, could achieve similar diagnostic accuracy compared to model A, with a complete four-phase protocol (accuracy 85.6% vs. 83.3%). Moreover, this work also shows that CT CNN models can improve accuracy, sensitivity and specificity compared to using CT alone (0.811–0.833, 0.744–0.923, and 0.725–0.941 vs. 0.543–0.676, 0.316–0.541 and 0.914, respectively). On a similar note, Yamada et al. [42] proposed transfer learning of pre-trained CNNs (GoogLeNet and Inception-v3) on a dataset of 215 patients with histologically proven primary liver cancers. Their approach consisted in splitting in a RGB system the results of different CT phases, obtaining in each similar accuracy to experienced abdominal radiologists and higher than general radiologists.

### 4.3. MRI

With the aim of an early-stage diagnosis and treatment to improve survival rates of liver cancer patients [46], MRI leads to a more comprehensive assessment of liver lesions and improved differential diagnosis, compared to CT. In addition, with the use of gadoxetic acid, MRI achieves a very high level of diagnostic accuracy in the diagnosis of HCC [49].

ML, and CNNs in particular, have also been applied to this modality, with promising results. Hamm et al. [43] employed a CNN to classify liver lesions on MRI, with an accuracy of 92%, a sensitivity of 92% and a specificity of 98% compared to a sensitivity and specificity of 82.5% and 96.5% obtained on these same cases by radiologists. Wu et al. [44] also investigated the usefulness of a CNN model for LI-RADS grading using a multiphase liver MRI. Specifically, the outcome of interest was the differentiation between LR-3 and LR-4/LR-5 tumors, achieving an accuracy of 90%, sensitivity of 100% and AUC of 0.95.

Interesting results were also obtained when information from MRI images was paired with clinical data. On this path, Jansen et al. [45] associated features extracted from DCE-MRI and T2-weighted sequences to risk factors and clinical data to create an automated classification system cataloguing liver lesions as adenoma, cyst, hemangioma, HCC and metastasis. Its sensitivity and specificity were 80/78%, 93/93%, 84/82%, 73/56% and 62/77%, respectively. Zhen et al. [46] developed a CNNs based on post contrast MRI, unenhanced MR images and clinical data which showed high accuracy in detecting and categorizing liver tumors. They were able to achieve a performance on par with three experienced radiologists for HCC diagnosis (AUC = 0.985; 95% CI = 0.960–1.000). Moreover, they highlight how images features united with clinical data and risk factors highly improved overall accuracy, compared to imaging alone, to classify HCC (AUC = 0.985; 95% CI = 0.960–1.000), metastatic tumors (AUC = 0.998; 95% CI = 0.989–1.000) and other primary malignancies (AUC = 0.963; 95% CI = 0.896–1.000). Agreement with pathology was 91.9%.

### 4.4. PET-CT

PET-CT has a proven role in staging and monitoring chemotherapy and radiotherapy outcome for several tumors. For HCC, it has no role as a first line diagnostic tool but can be useful for better characterization and follow-up, as well as for assessing prognosis [50]. In this setting, Preis et al. [47] employed a NN to analyze 18F-FDG PET-CT liver uptake of patient at risk of developing HCC, who also underwent MRI examined by expert radiologists. They report a high sensitivity and specificity for the detection of not yet identified liver malignancies, similar to performance of two expert radiologists on MRI (AUC = 0.896, 0.786 and 0.796, respectively).

## 5. Grading

The pathologic grade of HCC is a useful biomarker for survival and recurrence after surgery [51]. There are two classifications for HCC grade, both individuating four pathological grades based on cellular atypia and other factors. The World Health Organization classifies HCC in well-differentiated, moderately differentiated, poorly differentiated and undifferentiated, while Edmondson and Steiner classification identified grade I, II, III and IV with decreasing differentiation [51]. As with the determination of microvascular invasion (MVI), the preoperative assessment of HCC pathological grade is only possible by biopsy, which is not routinely recommended. ML may offer a noninvasive alternative for HCC grade prediction, as well as useful information regarding surgical resection margins or frequency of post-treatment follow-up.

Yang et al. [52] attempted to predict HCC pathological grade with a multichannel CNN. They extrapolated temporal sequence information and spatial texture information from 3D DCE-MRI of 42 HCC patients, correlating the dynamic pattern and structural morphology of HCC to pathologic grade. In this study, the CNN showed a diagnostic accuracy of 74% in the general differentiation of the HCC pathological. Specifically, its performance in discriminating well-differentiated HCCs from others was high, with AUC, accuracy, sensitivity and specificity of 0.96%, 91%, 96.88% and 89.62%, respectively. However, the performance for moderately differentiated and poorly differentiated HCC was lower. In order to find a relationship between imaging features and histological grading, another study [53] used texture analysis of DCE-MRI. In the end, two parameters, gray-level nonuniformity (GLN) and mean intensity, extracted from single arterial phase images were selected. Low-grade HCCs, compared with high-grade HCCs, showed an increase in mean intensity and a decreased GLN. In particular, AUC values of the average intensity and GLN were 0.918, 0.846, 0.836, 0.827 and 0.838, respectively. Wu et al. evaluated the diagnostic performance of non-contrast MRI radiomic signatures for the preoperative prediction of HCC grade. These signatures were created using the least absolute shrinkage and selection operator (LASSO) logistic regression model on features extracted from T1-weighted and T2-weighted images, alone and in combination. Their accuracy in predicting high grade and low-grade lesions was good (AUC = 0.712–0.742). In addition, of a combined clinical (age, sex, tumor size, alpha fetoprotein level, hepatitis B, hepatocirrhosis, portal vein thrombosis, portal hypertension and pseudo-capsule) and radiomics (using both sequences) model (AUC = 0.8) was better than that of a clinical (AUC = 0.6) or radiomic model alone (AUC = 0.7) [54]. Finally, another research study developed CECT-based radiomics signatures for preoperative prediction of HCC pathological grades using ML. These were based on arterial phase and on combined arterial and venous phase images, obtaining good accuracy in classifying high-grade HCC and low-grade HCC (AUC = 0.719 and 0.758, respectively). Again, the combination of the radiomics signature (built on arterial and venous phases) and clinical factors showed the best performance (AUC = 0.8) [55]. A summary of the reviewed articles is presented in Table 3.

## 6. Treatment Response and Prognosis Prediction

In clinical practice, the BCLC staging system is the most accepted among others and evaluates variables dependent both on the underlying chronic liver disease and neoplastic complication, allowing a prognostic stratification and proposing a therapeutic algorithm for HCC. However, current HCC guidelines have some limitations. The high variability in patient population, even within the same stage, as well as tumor heterogeneity, make prognostic assessment and therapeutic management of HCC very challenging. Therefore, AI techniques might provide additional, objective support in the decision-making process.

### 6.1. Microvascular Infiltration Prediction

MVI in HCC is recognized as an important predictor of poor survival and post-surgery tumor recurrence, negatively influencing the success of liver resection and transplantation [56]. Therefore, knowledge of its presence in the preoperative phase might influence decision making and allow more appropriate patient selection and surgical strategy.

However, unlike macrovascular invasion, MVI detection using imaging is problematic. Although different imaging findings, such as peritumoral enhancement, multifocality, irregular margins and capsular disruption, have been investigated as predictors, the results are still not reliable [56]. Currently, certain diagnosis is possible only through histopathological analysis after surgery, limiting the possible usefulness of MVI in actual practice.

Dong et al. trained a radiomics-based algorithm capable of predicting MVI-state and classifying MVI-positive patients in two groups, high and low risk. Radiomic signatures were based on features extracted from preoperative grayscale US imaging of 322 HCC cases through the analysis of tumoral and peri-tumoral region, as well as a combination of both. The results were promising with AUC values of 0.708, 0.710 and 0.726 for each. Furthermore, the addition of alfa fetoprotein values to the radiomic signatures further improved performance, achieving an AUC of 0.744.

The rationale for peri-tumoral area analysis is based on evidence that MVI occurs earlier in the tumor periphery rather than within the tumor, due to the increased expression of angiogenetic factors [57]. Another recently published research proposed a radiomic nomogram based on US-based features, alfa fetoprotein and tumor size. The resulting AUC for preoperative MVI identification was 0.731 [58].

Zheng et al. [59] extracted both tumoral and peri-tumoral area quantitative features from CECT and also reported that the association of clinical data improved the radiomic model, for larger tumors. They studied the association with preoperative clinical factors and qualitative radiographic descriptors by two radiologists. In particular, in patients with HCC ≤5 cm, quantitative features based on angle co-occurrence matrix had AUC of 0.8 in predicting MVI, while none of clinical factors or qualitative radiographic descriptors were correlated with MVI. In patients with tumors >5 cm a multivariate model combining AFP, tumor size, hepatitis status and quantitative feature had AUC of 0.88 in predicting MVI status.

Bakr et al. [60] evaluated computational features capturing lesion texture and shape from triphasic CECT as surrogate biomarkers of MVI, reaching an AUC of 0.76. Additionally, they proved diagnostic accuracy of their model was higher than previously reported signatures. Their best model combined single phase features and delta features in the arterial and venous phases, reaching AUC 0.76. Delta features were defined as the absolute difference and the ratio from all couples of phases for each feature.

Radiomic analysis of triphasic CECT to determinate textural variation within tumor lesions has also been investigated. The signature based on portal venous phase images had the best accuracy, further improved with a nomogram also integrating four clinical factors (AUC = 0.8) to predict MVI. It is evident that the integration of quantitative features, clinical and/or radiological qualitative biomarkers may provide added value to radiological scores in predicting MVI compared with radiomics models alone [61].

Another study [62] developed three radiomic models based on DCE-MRI using hepatobiliary-specific contrast agents. Again, radiomic features were extracted from both intra-tumoral and peri-tumoral regions with an AUC value of 0.83 for preoperative prediction of MVI.

Table 4 presents a summary of the reviewed articles.

### 6.2. Potentially Curative Therapies

Surgical resection, ablation and liver transplantation are considered potentially curative treatment for HCC in very early/early-stage disease (STAGE 0-A), according to guidelines [7]. Resection is the gold standard for patients with solitary tumors and well-preserved liver function while liver transplantation represents the optimal option for early-stage HCC patients meeting the Milano criteria, with poor functional liver reserves. However, only the 80% of all HCC cases can be treated through resection and transplantation is limited by restrictive selection criteria as well as availability of organs [63]. Thus, local ablation is an established therapeutic option in STAGE 0-A patients, who are not suitable for surgery or on a waiting list for liver transplantation, as well as selected BCLC STAGE B cases (combined with transarterial chemoembolization therapy, TACE). Recurrence of HCC after these therapies is still frequent, limiting their curative effect. Indeed, 5-year HCC recurrence affects 25% of patients after liver transplantation and more than 70% after hepatic resection or for radiofrequency ablation (RFA) [64]. Therefore, several ML models based on imaging data have been developed to predict treatment response, overall survival or risk of recurrence. These might provide information to better plan patient follow-up for high-risk patients or suggest alternative treatment strategies, such as a combination of therapies or bridge therapy before transplantation.

Zheng et al. [65] elaborated a radiomic score measured on arterial phase CECT of solitary HCC patients assigned to liver resection, using a LASSO regression model to predict post-operative survival and recurrence. A low radiomics score was significantly associated with shorter postoperative survival and recurrence. Indeed, patients with low score tended to present features suggestive of aggressive cancer such as high alfa fetoprotein levels, larger tumor size, vascular invasion and advanced stage. A radiomics-based nomogram had good predictive accuracy of survival (C-index = 0.71). Further, after adding TNM and BCLC stage, the C-index further increased, suggesting that the model might be complementary to traditional staging systems. A multi-institutional study [66] tested a radiomics model based on CECT analysis to predict recurrence of HCC after liver resection. Features to train the ML algorithms were extracted and selected from the lesion and its periphery. Overall, two ML-based radiomics models were designed. The first was a preoperative model was based on a radiomic signature and parameters available before surgery (alfa fetoprotein, albumin-bilirubin grade and liver cirrhosis); the second, post-operative model included both the preoperative data and pathological results (tumor margin and satellite nodules). Both models showed higher prognostic performance compared to other non-radiomic models and current staging system, with C-index = 0.733–0.801. Furthermore, these models provide three risk categories with increasing probability of recurrence and distinct recurrence patterns that might influence the surgical strategy and use of adjuvant therapies as well as personalized surveillance policy.

A recent paper by Yuan et al. [67] had the assessment of recurrence free survival (RFS) after curative ablation as its endpoint. Several radiomic features were extracted from triphasic CECT images of 184 HCC patients assigned to ablation therapy. Among these, 20 were selected and used as input to the LASSO Cox model to generate a radiomics signature. Moreover, two categories of risk, low and high, were identified based on the resulting model. Among the other, portal venous phase radiomic model had the higher performance (C-index = 0.736) and the combined model integrating these radiomic signatures and clinicopathologic features increasing predictive performance (C-index = 0.755). Moreover, it was built a nomogram based on this final model, capable of assess the 1-, 2- and 3-year RFS rates.

Recurrence prediction for HCC after liver transplantation, as the main factor that influence its curative efficacy, was the aim of another study by Guo et al. [68], where they developed a radiomics signature through features extracted from CECT arterial phase using a LASSO Cox model. This was again combined with clinical risk factors to predict RFS with a C-index of 0.789, also being able to stratify patients into high- and low-risk categories.

Another research area is represented by the development of models that could aid the choice between liver resection or tumor ablation. Liu et al. [69] developed a DL-based radiomics strategy using a CNN model to extract space-time features from CEUS time-intensity curves in early and very early-HCC patients, undergoing either surgical resection or RFA. Two radiomics model and two nomograms built by incorporating radiomic signatures and clinical data had good predictive performance for progression-free survival in both patient groups (C-index = 0.726 for RFA, 0.741 for resection), proving that their approach might be helpful in selection of the optimal therapeutic strategy between the two types of treatment.

Researchers have also tried to use a radiomics approach based on DCE-MRI for prevision of pre-operative recurrence or survival in HCC patients after resection and/or ablation. Zhang et al. [70] developed radiomic signatures based on preoperative DCE-MRI using hepatobiliary-specific contrast agents to predict overall survival in surgically resected HCC patients. The tumor, its periphery and non-tumoral parenchyma were quantitatively analyzed to build the models. As a result, three radiomics scores, based on three different regions of interest (tumor, perilesional and non-tumoral parenchyma) were obtained. Among these, the non-tumoral parenchyma score had the highest prognostic performance (C-index = 0.72), proving the importance of liver background as a prognostic factor. Moreover, the combined score, derived from all three regions of interest, had a C-index of 0.83. The association with clinical-radiological predictors (BCLC stage; non-smooth tumor margins) provided the final model with the best prognostic performance for survival outcome (C-index 0.84).

Finally, unlike most models which are based on preoperative imaging, the ML-radiomics one proposed by Shen et al. [71] was trained on post-treatment CECT exams (within one month after resection or ablation), analyzing lesions suspect for HCC recurrence (<2 cm) without classical dynamic features. They aimed to improve early detection of recurrence after therapy to allow more timely treatment. Among those extracted, 34 features significantly correlated to recurrence were selected by a random forest method. The resulting model had better performance than alfa fetoprotein levels in early detection of recurrence (AUC = 0.89 and 0.83, respectively). The combination of both did not significantly improve performance, supporting the model’s potential role in HCC patient follow-up after resection or RFA. Specifically, in the validation cohort, for the radiomics model the AUC, accuracy, sensitivity, specificity, negative and positive predictive values in detecting recurrence were 0.89, 0.86, 0.91, 0.75, 0.82 and 0.88.

In Table 5, a summary of the papers considering potentially curative therapies response is presented.

### 6.3. Trans-Arterial Therapies

TACE represents the first therapeutical approach for intermediate stage HCC (stage B) in patients with unresectable tumors. However, local HCC response to this therapy is extremely variable. Therefore, prediction of response to TACE could be very helpful for treatment planning and help in selecting patients who can benefit the most from repeated TACE procedures to induce tumor complete necrosis or individuate non-responders that may take advantage of alternative strategies. Moreover, unnecessary TACE procedures can lead to adverse reactions, including serious ones, as well as increased healthcare costs. Subclassifications of stage B and scoring systems based on clinical, radiological and biological data, such as the hepatoma arterial-embolization prognostic and selection for trans-arterial chemoembolization treatment scores, have been proposed but still present limited predictive accuracy [72,73,74]. Recently, the potential of radiomics, AI and ML applied to pre-procedural imaging also has drawn interest in identification of potential predictors of response to TACE.

Peng et al. [75] trained a CNN to predict the response to TACE using CT images of a total of 789 patients from three different centers. Their model had an accuracy of 84.3% and AUCs of 0.97, 0.96, 0.95 and 0.96 for complete response, partial response, stable disease and progressive disease prediction. In another study (pre-TACE tumor signal intensity) [76], a combination of baseline MRI and clinical features were used to train two ML models (logistic regression and random forest) to classify patients as TACE responders or non-responders. The best overall accuracy was 78% (sensitivity = 62.5%, specificity = 82.1%, positive predictive value = 50% and negative predictive value = 88.5%), reached by training models with two features, one clinical (presence of cirrhosis) and one imaging-derived.

Researchers [77] also investigated the potential of CEUS and B-mode US in predicting patient response to TACE. Specifically, a 3D CNN based on CEUS was constructed to analyze quantitatively pre-procedural dynamic CEUS images of 130 HCC patients with BCLC stage B undergoing TACE. This NN had better predictive performance (AUC = 0.93) than other two ML radiomics models built in this study which respectively analyzed CEUS time-intensity curve and B-mode US. Furthermore, the 3D CNN proved to be superior compared to the hepatoma arterial-embolization prognostic score in predicting TACE outcome.

Some investigations focused the contribution of textural analysis and DL, respectively, in selecting patients who can benefit from simultaneous association of TACE and sorafenib administration [78,79]. This combination is based on the inhibition of VEGF up-regulation induced by TACE, but its efficacy is still controversial. Zhang and colleagues [79] validated a CNN model to predict overall survival from preoperative CE CT images of advanced HCC patients treated with TACE and sorafenib. They built an integrated nomogram based on clinical features and a DL-signature. Their results were promising, with a high prediction performance of the NN (C-index = 0.717) and of the combined nomogram, which proved to be better than the clinical nomogram (C-index = 0.730 and 0.679).

Another locoregional treatment option, acquiring popularity as an alternative to TACE, is trans-arterial radioembolization with Yttrium-90 (^90^Y-TARE). Blanc-Durand et al. [80] evaluated the pretreatment ^18^F-FDG PET of 47 patients undergoing this procedure, extracting quantitative features by texture and intensity analyses of the entire liver. The resulting predictive radiomics score, obtained using LASSO regression, showed significant correlation with progression-free and overall survival, stratifying patients in low-risk and high-risk subgroups.

An overview of these articles is presented in Table 6.

## 7. Limitations and Future Perspectives

While recent years have seen exponential growth of radiomics, AI and ML applications in HCC imaging, as in other fields, there are still several limitations that should be addressed prior to the introduction of these tools in clinical practice. Most studies in this field are retrospectively designed, which leads to potential selection biases in patient populations [81]. The data quality and limited nature of currently employed datasets in relation to model complexity, especially for NNs, or small sample size open the door to the risk of overfitting, i.e., the excessive tailoring of algorithms to training data [8,82]. Thus, it would be desirable to perform future studies on larger patient populations collected with prospective multicenter trials to obtain more reliable results. This approach would also solve the issue represented by lack of external validation of the results in many studies on an independently collected population [82,83,84]. Availability of high-quality public databases including imaging, histopathologic and genomic data could be a solution, allowing researchers to reproduce and replicate their results on larger scale [85]. However, these are challenging to collect and share, and still require quality checks to ensure their validity and representativeness of the general population over time [86,87]. Moreover, the lack of standardization in AI pipelines, imaging acquisition parameters and segmentation methods also negatively impact reproducibility and comparability between studies. Greater consistency in reporting methods and results in this type of research would also help the field’s growth [88,89]. From a practical perspective, another issue that hinders adoption of ML in healthcare is represented by the allocation of responsibility for diagnostic errors, as well as the potentially exponential effect of an incorrect model’s use compared to a single doctor-patient interaction [88]. This issue is further compounded by the lack of interpretability found in many ML models, especially DL solutions [88,90].

Despite these challenges, as highlighted in our review, the potential of AI techniques is huge in each phase of HCC management, ranging from initial diagnosis to treatment selection and prognostic and therapeutic response prediction. These tools could further the evolution towards precision and personalized medicine to support clinical practice and optimize costs and resources [11,15,81,89]. It should also be noted that the results from several investigations support the integration of the ML models with clinical-pathological data and established clinical scores or biomarkers.

## 8. Conclusions

ML technology and radiomics are still immature and require several improvements before clinical implementation in management of HCC patients. Nevertheless, the awareness of their potential is the premise for improving research standards and incentivize their adoption by the scientific community as a tool that will support and enhance human intelligence without replacing it.

## Figures and Tables

**Figure 1 diagnostics-11-01194-f001:**
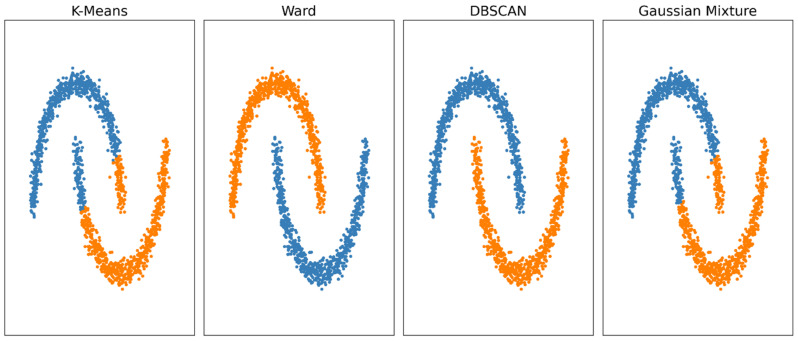
Output of various unsupervised clustering models (k-means, ward, DBSCAN and gaussian mixture) on a 2-feature synthetic dataset with a noisy half-moon morphology for each cluster. The color-coded output depicts the performance of each model in relation to this dataset’s structure, demonstrating how some approaches may be better suited than others.

**Table 1 diagnostics-11-01194-t001:** Overview of ML algorithms applied to lesion segmentation.

Authors	ML Algorithm	Aim	Imaging Modality	Performance
Christ et al. [20]	Cascaded CNNs based on a U-Net Architecture	Liver and tumor segmentation	CT	DICE scores: 94.3% and 91% *
Ouhmich et al. [25]	Cascaded CNNs based on a U-Net Architecture	Segmentation of healthy and cancerous liver tissues, discriminating normal parenchyma, active HCC and necrotic tumoral tissue	CT	DICE score: 90.5%, 59.6% and 75.8% **
Zhang et al. [26]	Auto-context-based CNNs based on a U-Net Architecture	Liver Tissue Classification	MRI	F-Score: 0.80, 0.83 and 0.81 ***
Han et al. [27]	CNN based on 2.5D model	Liver lesion segmentation	CT	DICE score: 0,67
Wardhana et al. [29]	CNN based on 2.5D model	Liver and lesion segmentation	CT	Lesion DICE score: 78.4 ± 16.7 and 83.6 ± 24.7% **** Liver DICE score: 95.3 ± 1.8% and 94.6 ± 2% ****
Conze PH. et al. [30]	Scale-adaptive supervoxel-based random forests	Liver tumor segmentation	CT	TN rate error of Δτ: 4.08. DSC: p95.9, a80.3, n86, t92.6 *****
Chih-Yu Hsu et al. [31]	Poisson Gradient Vector Flow-ACM based on a genetic algorithm	Liver segmentation	PET	Reduction of the iterations needed in liver’s edge processing selection

ML: machine learning; CNN: convolutional neural network; ACM: active contour model; AUC: area under the curve; TN: tumor necrosis; DSC: dice similarity coefficients. * Results respectively on 3DIRCAD (3D Image Reconstruction for Comparison of Algorithm Database) dataset and Clinical CT dataset. ** Results respectively for parenchyma, active tumor and necrosis. *** Results respectively by multi-resolution input multi-phase training, multi-resolution input single-phase training and single-resolution input single-phase training. **** Results respectively by Net01 and Net02. ***** *p*: parenchyma; a: active; n: necrosis; t: tumoral.

**Table 2 diagnostics-11-01194-t002:** Overview of ML algorithms applied to hepatocellular carcinoma diagnosis.

Authors	ML Algorithm	Aim	Imaging Modality	Performance
Bharti et al. [33]	CNN based on ensemble model (k-NN, SVM and rotation forest)	Classify four classes of liver images on US, namely normal liver, chronic liver disease, cirrhosis and HCC	US	Accuracy: 96.6%
Schmauch et al. [34]	ResNet50 Neural Network	Detect and classify liver lesions as benign or ultrasound malignant	US	AUC: 0.93 and 0.91 *
Hassan et al. [35]	stacked sparse auto-encoder with SoftMax layer classifier	Detect HCC, hemangioma and liver cysts	US	Sensitivity: 98% Specificity: 95.7%
Guo et al. [36]	deep canonical correlation analysis-multiple kernel learning based classifier	Discriminate benign and malignant liver lesions	CEUS	Accuracy: 90.41 ± 5.80%
Mokrane et al. [37]	k-NN, SVM and RF	Classify hepatic nodules as HCC or non-HCC	CT	AUC: 0.66
Yasaka et al. [38]	CNN	Classification of liver lesions in five categories	CT	AUC: 0.92
Raman et al. [39]	RF	Classification of hypervascular liver lesions	CT	Accuracy: 90%
Nayak et al. [32]	SVM	Diagnosis of cirrhosis and hepatocellular carcinoma	CT	DICE score: 90%, 86% and 81% **
Vivanti et al. [40]	CNN	Detection of tumor recurrence based on CT volume/tumor load	CT	Accuracy: 86%
Wenqi et al. [41]	CNN	Diagnostic accuracy of a three-phase CT protocol without PV vs. four-phase CT protocol	CT	Accuracy: 85.6% vs. 83.3%
Yamada et al. [42]	CNN	Diagnosis of primary liver cancers using transfer learning	CT	Mean DP: 44.1%, 44.2%, and 43.7% ***
Hamm et al. [43]	CNN	Classify liver lesions	MRI	Accuracy: 92%
Wu et al. [44]	CNN	CNN model for LI-RADS grading	MRI	AUC: 0,95 ****
Jansen et al. [45]	extremely randomized trees classifier	Automated classification system cataloguing liver lesions as adenoma, cyst, hemangioma, HCC and metastasis	MRI	Sensitivity/Specificity: 80/78%, 93/93%, 84/82%, 73/56% and 62/77% *****
Zhen et al. [46]	CNN	Detecting and categorizing liver tumors	MRI	AUC: 0.98, 0.99, 0.96 ******
Preis et al. [47]	ANN	Analyze 18F-FDG PET-CT liver uptake of patient at risk of developing HCC	PET	AUC: 0.89

ML: machine learning; CNN: convolutional neural network; k-NN: k-nearest neighbor; SVM: support vector machine; RF: random forest; DL: deep learning; PV: portal venous; AUC: area under the curve; DP: diagnostic performance; ANN: artificial neural network. * Results respectively for focal liver lesion detection and focal liver lesion characterization. ** Results respectively for healthy liver, cirrhosis and HCC. *** Results respectively by pixel shifts, rotations, and skew misalignments transfer learning methods. **** Result for differentiation between LR-3 and LR-4/LR-5 tumors. ***** Results respectively for adenoma, cyst, hemangioma, HCC and metastasis detection. ****** Results respectively for HCC, metastasis and other primary malignancies.

**Table 3 diagnostics-11-01194-t003:** Overview of ML algorithms applied to hepatocellular carcinoma grade prediction.

Authors	ML Algorithm	Aim	Imaging Modality	Performance
Yang et al. [52]	DL MCF-3D CNN	Prediction of grade	DCE MRI	Differentiation of grades: accuracy 74% WD HCC: AUC 0.96; accuracy 0.91; sen 97%; spe 89%
Wu et al. [54]	Logistic regression	Prediction of grade	non-contrast-enhanced MRI	Combined T1WI and T2WI-based radiomics signatures: AUC 0.74 Combined radiomics signatures + clinical factors: AUC 0.8
Mao et al. [55]	XGBoost	Prediction of grade	CE CT	Combined AP and VP-based radiomics signatures: AUC 0.75 Combined radiomic signature (built on AP and VP) + clinical factors: AUC 0.8

AP: arterial phase; AUC: area under the curve; CE CT: contrast enhanced computed tomography; CE MRI; DCE MRI: dynamic contrast enhanced magnetic resonance imaging; DL MCF-3D CNN: deep-learning multichannel fusion three-dimensional convolutional neural network; ML machine learning; sen: sensibility; spe: specificity; XGBoost: extreme gradient boosting; T1WI: T1 weighted imaging; T2WI: T2 weighted imaging; VP: venous phase; WD HCC: well-differentiated hepatocellular carcinoma.

**Table 4 diagnostics-11-01194-t004:** Overview of ML algorithms applied to microvascular invasion prediction.

Authors	ML Algorithm	Aim	Imaging Modality	Performance
Dong et al. [57]	Logistic regression	Prediction of MVI	US	TR, PTR, and combined TR-PTR-radiomic models: AUC 0.708; 0.71 and 0.726 Combined TR-PTR signatures + AFP values: AUC 0.744
Bakr et al. [60]	Lasso	Prediction of MVI	CECT	Combined model based on single phase features + arterial-venous delta features: AUC 0.76
Ma et al. [61]	SVM	Prediction of MVI	CECT	PVP radiomic signature: AUC 0.79 PVP radiomic signature + clinical factors: AUC 0.8
Feng et al. [62]	Logistic regression	Prediction of MVI	DCE MRI	Combined intra-tumoral and peri-tumoral radiomics model: AUC 0.83

AFP: alpha fetoprotein; AUC: area under the curve; CECT: contrast enhanced computed tomography; DCE MRI: dynamic contrast enhanced magnetic resonance imaging; ML: machine learning; MVI: microvascular invasion; PTR: peri-tumoral region; PVP: portal venous phase; TR: tumoral region; US: ultrasound; SVM: support vector machine.

**Table 5 diagnostics-11-01194-t005:** Overview of ML algorithms applied to prediction of potentially curative therapies response.

Authors	ML Algorithm	Aim	Imaging Modality	Performance
Ji et al. [66]	Unsupervised clustering analysis	Prediction of recurrence after SR	CECT	Pre-operative model: C-index 0.733 Post-operative model: C-index 0.801
Yuan et al. [67]	Lasso	Prediction of RFS of HCC after SR	CECT	PVP radiomic model: C-index 0.736 Combined model based on clinicopathologic features + PVP radiomic signature: C-index 0.755
Guo et al. [68]	Lasso	Identification of aggressive behavior of HCC and prediction of HCC RFS after liver transplantation	CECT	AP radiomic model: C-index 0.705 Combined model based on AP radiomic signature+ clinical risk factors C-index 0.789
Liu et al. [69]	CNN	Prediction of PFS of RFA and SR and optimize the treatment selection in very-early and early-stage HCC	CEUS	Radiomic model RFA: C-index 0.726 Radiomic model SR: C-index 0.741
Zhang et al. [70]	Lasso	Prediction of OS after SR	CE MRI	Non-tumoral parenchyma-score: C-index 0.72 Combined Rad-score (from 3 ROI): C-index 0.83 Combined model based on Rad-score + clinical-radiological predictors: C-index 0.84
Shen et al. [71]	Random forest	To improve the performance of detecting recurrence after therapy to allow for an early strategy	CECT	Radiomic model: AUC 0.89 Combined model based on radiomic algorithm + chance of AFP: AUC 0.89

AFP: alpha fetoprotein; AP: arterial phase; CECT: contrast enhanced computed tomography; CEUS: contrast enhanced ultrasound; DL CNN: deep learning convolutional neural network; OS: overall survival; PFS: progression-free survival; PVP: portal venous phase; RFA: radiofrequency ablation; RFS: recurrence free survival; ROI: region of interest; SR: surgical resection.

**Table 6 diagnostics-11-01194-t006:** Overview of ML algorithms applied to trans-arterial therapies response prediction.

Authors	ML Algorithm	Aim of Study	Imaging Modality	Performance
Peng et al. [75]	CNN	Prediction of response to TACE	CE CT	Accuracy 83.3%; CR, PR, SD, PD: AUCs 0.97, 0.96, 0.95 and 0.96
Abajian et al. [76]	Logistic regression, random forest	Prediction of response to TACE	CE MRI	RF and LR models: overall accuracy 78%
Liu et al. [77]	CNN	Prediction of response to TACE	B-mode US, CEUS	R-DLCEUS model: AUC 0.93 R-TIC model: AUC 0.8 R-BMode model: AUC 0.78
Zhang et al. [79]	CNN	Prediction of OS in patients treated with TACE + sorafenib	CECT	DL-model: AUC 0.717 Combined nomogram based on DL-signature + clinical features: AUC 0.73
Blanc-Durand et al. [80]	Lasso	Prediction of OS and PSF after ^90^Y-TARE	18F-FDG PET	PFS-PET-RadScore (*P* = 0.006) and OS-PET-RadScore (*P* = 0.001) were independent negative predictors for PFS and OS respectively

CE CT: contrast enhanced computed tomography; CEUS: contrast enhanced ultrasound; CR: complete response; DL CNN: deep learning convolutional neural network; ^18^F-FDG PET: fluorodeoxyglucose positron emission tomography; ML: machine learning; OS: overall survival; PD; progressive disease; PFS: progression-free survival; PR: partial response; SD: stable disease; R-BMode: radiomics-based B-mode images; R-DLCEUS: radiomics-based-deep learning contrast enhanced ultrasound; R-TIC: radiomics-based time intensity curve of CEUS; TACE: trans-arterial chemoembolization; ^90^Y-TARE: transarterial radioembolization using Yttrium-90.

## Data Availability

Not applicable.

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
