# Peer review of "State of the Art in Artificial Intelligence and Radiomics in Hepatocellular Carcinoma"

_diagnostics, 2021, doi:10.3390/diagnostics11071194_

Round 1
Reviewer 1 Report
It is a well-written review article, easy to follow and enjoyable to read.
The Authors should consider the following (minor) revisions:
- Figure 1 is no necessary as it hardly contributes to the information content of the article. In my opinion, it can be removed without any loss for the article.
- Please, go through the text once again to check English as there are a few mistakes and typos.
Reviewer 2 Report
A good overview of the ways AI can be employed in HCC imaging, this is work is of great importance in the age of non-invasive diagnosis of HCC. Minor point, please cite articles appropriately. e.g. [1] rather than 1, at the end of each statement.
Reviewer 3 Report
General Comments:
The review aims to describe the main results and future perspectives of Artificial Intelligence (AI) and its subfields in early diagnosis, prognosis and choice of proper treatment of hepatocellular carcinoma (HCC). In fact, conventional and morphological imaging has relevant limitations in HCC management, especially in HCC diagnosis and prediction prognosis. These issues could affect patient prognosis and treatment options. Then, AI could represent the future imaging landscape in these patients with the proposal to enter in HCC workup.
The object of the paper is very interesting; however, the manuscript has several limitations in almost all sections.
Please see specific comments.
Specific comments:
Major weak points:
- Please consider adding a dedicate section concerning technical principals, in this manner you can provide a detailed overview of AI, its subfields, and the main steps of image analysis from image acquisition to model building.
- Please add the “Conclusion section”
- Please add some references that seems to be lacking in the paper.
- Please spell out each abbreviation the first time that you mention.
- Please add the “Abbreviation list”.
Title: Please remove “Radiomics” from the title.
Keywords: You might consider replacing “Radiology” with “Imaging”.
Abstract:
- Please consider avoiding some details concerning mortality and screening of HCC in the first paragraph, and expanding in a more detailed and precise manner the main imaging challenges.
- Please avoid “lesion segmentation” in the range of radiomics and ML applications.
Background:
- You should consider avoiding some epidemiological HCC aspects, that seem to be unnecessary in this review. Please rewrite the section in synthetic manner.
- Please describe the literature data regarding the accuracy, sensibility and specificity of each method of imaging in HCC diagnosis. Please specify the imaging method considered as goal standard according to HCC international guidelines.
- Please describe in a more detailed manner the main issue of imaging about HCC management, from diagnosis to prognosis prediction. Could AI routinely support the clinicians?
- What about some previous studies which investigated the role of radiomics and ML applied to HCC? Please expand this aspect.
- Please describe the future role of radiomics and ML in HCC diagnosis, prognosis and response to therapy evaluation, these could solve some actual imaging challenges?
- What about the main advantages and drawbacks of AI? Please discuss.
- Please specify the main future perspectives of radiomics and ML in HCC workup.
- Please define in a more precise manner the aim of the study, it seems to be quite inconsistent. Please avoid “lesion segmentation” among the main AI applications.
- Please add some references, these seem to be lacking especially in the first paragraph.
Radiomics and Machine Learning:
- Please modify the title, it seems to be reductive considering that in this section you describe also Deep Learning approach.
- Please rewrite the section in a more synthetic and punctual manner. This seems to be overall confusing and difficult to follow.
- P3L115-119: Please avoid some general expression without a proper literature discussion.
- Please be more consistent in the description of practical application of each AI subfields.
- Please add some references.
Lesion Segmentation:
- Please be more consistent in the description of main strengths and drawbacks for each type of segmentation. several methods have been proposed to overcome the main limitations of tumor segmentation, please describe.
- Please describe the type of tumor segmentation for each study that you mentioned. In addition, you should describe the main results obtained.
- You should provide an overview of all methods of lesion segmentation and describe the promising results obtained from major studies, with the aim to render the method stable and reliable in the clinical practice. The section seems to be overall confusing.
Diagnosis:
- Overall, please rewrite the section which seems to be confusing. You should describe the main issues for each method of Imaging (i.e. US, CT, MRI) and describe the adding role of AI in clinical practice.
- Please describe the key points of each study that you described, the method and results. You should structure each subparagraph according to screening and diagnosis of liver nodules.
- What about LI-RADS criteria? Please define the application and if you can use in all cirrhotic patients.
- You should introduce each imaging challenges before the description of several studies, please modify.
- Please consider to expand or to avoid the subparagraph “PET/CT”, it seems to be inconsistent.
Grading:
- Please briefly describe the main limitations of conventional imaging in the assessment of HCC grading and MVI. You should mention MVI in the dedicated section.
- Please rewrite the section in a more structured manner, you should avoid some general expression such as “another research study” and “again”, these seem to be unappropriated in this paper.
- Please consider to describe in a more concise manner each study, focusing on the main results.
- Please add some references.
Treatment response and prognosis prediction:
- Please expand the section, this seems to be inconsistent.
Microvascular infiltration prediction:
- What about the use of conventional MRI on detection of MVI? Please briefly describe some literature data about sensitivity and specificity.
Potentially curative therapies
- Please modify the first paragraph concerning HCC therapy guidelines, you might avoid some non-relevant information.
- Please introduce in a proper manner the first group of papers that you decided to mention (L413-449), you should enhance the aim of each field of AI application.
Trans-arterial therapies
- Please rewrite in a more synthetic manner the section and consider introducing the aim of the paper discussed. Sometimes, the section seems to be fragmentary.
Limitations and future perspectives
- Please rewrite the section in a more synthetic manner, it seems overall confusing.
- Please specify the main limitations of AI in a more structured manner, the description sounds quite unclear. You should describe the main limitation concerning each step of quantitative analysis from image acquisition to modelling. Please modify.
References: Please see comments above and modify the style according to diagnostics guidelines.
Tables: overall ok
Figures:
- Figure1: This seems to be not relevant according to the aim of the paper.
- Figure 2: it seems to be lacking, please check.
Linguistic and typewriting: English writing needs some important improvements.
Round 2
Reviewer 3 Report
Dear Authors,
thanks for the changes in the draft.
Unfortunately not all comments have been replied.
Please at least improve the background, radiomics and machine learning, lesion segmentation paragraphs, which are still lacking of relevant information.
